# SGLT-2 Inhibitors and Metabolic Outcomes: A Primary Data Study Exploring the Microbiota–Diabetes Connection

**DOI:** 10.3390/metabo15060411

**Published:** 2025-06-18

**Authors:** Nicoleta Mihaela Mindrescu, Cristian Guja, Viorel Jinga, Sorina Ispas, Antoanela Curici, Rucsandra Elena Danciulescu Miulescu, Andreea Nelson Twakor, Anca Mihaela Pantea Stoian

**Affiliations:** 1Faculty of Medicine and Pharmacy, “Carol Davila” University, 050474 Bucharest, Romania; nicoleta-mihaela.mindrescu@drd.umfcd.ro (N.M.M.); viorel.jinga@umfcd.ro (V.J.); rucsandra.danciulescu@umfcd.ro (R.E.D.M.); anca.stoian@umfcd.ro (A.M.P.S.); 2National Institute of Diabetes “NC Paulescu”, 030167 Bucharest, Romania; 3Department of Urology, Clinical Hospital “Prof. Dr. Th. Burghele”, 061344 Bucharest, Romania; 4Department of Anatomy, Faculty of General Medicine, “Ovidius” University, 900470 Constanta, Romania; sorina.ispas@365.univ-ovidius.ro; 5Department of Cellular and Molecular Biology, and Histology, “Carol Davila” University of Medicine and Pharmacy, 050474 Bucharest, Romania; antoanela.curici@umfcd.ro; 6Department of Internal Medicine, Emergency County Hospital, 900591 Constanta, Romania

**Keywords:** gut microbiota, type 2 diabetes mellitus, lifestyle factors, dietary habits, sedentary behavior, chronic stress, metabolic risk, dysbiosis, insulin resistance

## Abstract

Background: The gut microbiota plays a critical role in metabolic health and type 2 diabetes mellitus (T2DM). Alterations in microbial composition may influence glycemic control and systemic inflammation. Materials and methods: In this single-center, randomized study, 60 adults with T2DM receiving metformin were evaluated biologically and received either empagliflozin or sitagliptin. Demographic, metabolic, and lifestyle data were collected. Gut microbiota profiling was conducted at two timepoints to assess changes in bacterial and fungal taxa. Blood glucose, HbA1c, and inflammation markers were analyzed longitudinally. Results: Both treatment groups showed significant improvements in glycemic control. Median fasting glucose decreased from 132 to 123 mg/dL (*p* = 0.046) in the sitagliptin group and from 131 to 114 mg/dL (*p* = 0.025) in the empagliflozin group. Median HbA1c levels declined significantly in both groups, with a greater reduction in the empagliflozin group (*p* = 0.001 vs. *p* = 0.049). The microbiota analysis revealed an increase in beneficial bacteria (e.g., *Bifidobacterium* spp. and *Lactobacillus* spp.) and a decrease in pro-inflammatory taxa *(Escherichia coli* and *Streptococcus* spp.). Notably, empagliflozin was associated with a more pronounced microbiota rebalancing and a significant decline in fungal overgrowth (e.g., *Candida* spp.; *p* = 0.034). Conclusions: Treatment with sitagliptin and empagliflozin led to improved glycemic outcomes and partial restoration of gut microbial balance in T2DM patients. Empagliflozin showed superior efficacy in modulating both glycemia and dysbiosis.

## 1. Introduction

The human gastrointestinal tract harbors a complex and dynamic community of microorganisms, collectively known as the gut microbiota. This microbial ecosystem plays a pivotal role in various physiological processes, including digestion, immune modulation, and metabolic regulation [1]. Recent research has illuminated the significant influence of gut microbiota on the development and progression of metabolic disorders, notably type 2 diabetes mellitus.

Located primarily in the intestines, the gut microbiota influences not only local digestive functions but also distant organs such as the brain, liver, heart, and skin [2]. Through its interaction with the immune system, metabolic pathways, and the enteric nervous system, the microbiota can contribute to a wide spectrum of conditions ranging from neurodegenerative diseases (like Parkinson’s and Alzheimer’s) to metabolic disorders including diabetes, obesity, and insulin resistance [3]. It also plays a critical role in modulating emotional health, with links to depression, anxiety, stress, and even addiction—underscoring the gut–brain axis.

Figure 1 highlights the role of dysbiosis (microbial imbalance) in chronic inflammation, liver and cardiovascular disease, and autoimmune conditions such as eczema and arthritis. These connections emphasize that gut microbes are not isolated to gastrointestinal health but are fundamental to whole-body physiology. The bidirectional arrows reflect the complex feedback loops where microbiota composition both influences and is influenced by host health, diet, environment, and behavior.

One of the most important first-line treatments for type 2 diabetes is metformin [5]. It has many pleiotropic impacts on systems and processes in addition to being an antihyperglycemic drug. AMPK activation in cells and hepatic glucose reduction are its main effects [6]. It reduces endothelial advanced glycation end products and reactive oxygen species formation and regulates cardiomyocyte glucose and lipid metabolism, reducing cardiovascular risks [7].

Sodium-glucose co-transporter type 2 inhibition is a novel diabetic treatment that has garnered significant interest. For the first time, an insulin-independent approach achieves glucose-lowering effects by targeting an organ that is crucial to glucose metabolism but has been overlooked in T2DM medication development [8].

SGLT2 inhibitors gained popularity after the EMPA-REG OUTCOME research revealed a reduced risk of major CV events or mortality in high-risk T2DM patients with empagliflozin added to their conventional treatment [9].

DPP-4 inhibitors have antidiabetic effects by boosting insulin production through selective inhibition of the enzyme, which inactivates incretins such glucagon-like peptide 1 and stomach inhibitory polypeptide by a separate mechanism from traditional hypoglycemics [10]. Many studies [11,12] have shown the better efficacy and safety of DPP-4 inhibitors, which was initially approved in Japan in 2009 [13].

T2DM is characterized by chronic hyperglycemia resulting from insulin resistance and impaired insulin secretion [5]. While genetic predisposition contributes to T2DM risk, environmental factors, particularly diet and lifestyle, are critical determinants. Emerging evidence suggests that alterations in gut microbiota composition—termed dysbiosis—may be a key intermediary linking lifestyle factors to metabolic dysfunction [8].

As it can be seen in Figure 2 below, one of the primary mechanisms by which gut microbiota influence host metabolism is through the fermentation of dietary fibers into short-chain fatty acids (SCFAs), such as acetate, propionate, and butyrate [3]

These SCFAs serve as energy sources and signaling molecules, modulating glucose and lipid metabolism, enhancing insulin sensitivity, and exerting anti-inflammatory effects. For instance, butyrate has been shown to improve insulin sensitivity and reduce inflammation in adipose tissue [6].

Conversely, dysbiosis can lead to increased intestinal permeability, facilitating the translocation of lipopolysaccharides (LPS) into systemic circulation—a condition known as metabolic endotoxemia [2]. Elevated LPS levels trigger chronic low-grade inflammation, a hallmark of insulin resistance and T2DM. Studies have demonstrated that individuals with T2DM exhibit higher circulating LPS levels compared to healthy controls, correlating with markers of inflammation and insulin resistance [5].

Moreover, specific bacterial taxa have been associated with T2DM. A decrease in beneficial butyrate-producing bacteria, such as *Faecalibacterium prausnitzii* and *Roseburia* spp., alongside an increase in opportunistic pathogens like *Ralstonia pickettii*, has been observed in diabetic individuals [11]. These microbial shifts may disrupt SCFA production and promote inflammatory pathways, exacerbating metabolic disturbances [12].

Interventions targeting the gut microbiota have shown promise in modulating metabolic outcomes. Probiotic and prebiotic supplementation can restore microbial balance, enhance SCFA production, and improve glycemic control [13]. Additionally, dietary modifications emphasizing high-fiber intake support the growth of beneficial microbes, reinforcing gut barrier integrity and reducing inflammation.

Pharmacological agents, notably metformin, also interact with the gut microbiota. Metformin has been found to alter microbial composition, increasing the abundance of *Akkermansia muciniphila*, a bacterium associated with improved metabolic profiles [10].

To strengthen the novelty of this work, our study stands apart from previous research by focusing on a primary dataset generated specifically for this analysis, rather than relying on secondary or pooled sources. Importantly, we offer a direct head-to-head comparison between two widely used antidiabetic drug classes—SGLT-2 inhibitors (empagliflozin) and DPP-4 inhibitors (sitagliptin)—within a real-world type 2 diabetes population. Furthermore, we combine clinical and biochemical outcomes with a simplified PCR-based gut microbiota profiling approach, providing an integrated perspective on how pharmacologic interventions interact with microbial patterns.


Top of Form



Bottom of Form


## 2. Materials and Methods

This single-center study examined the gut microbiota, lifestyle, and metabolic health of 60 patients recruited between January 2023 and December 2024. The study protocol was approved by the Ethics Committee of the “N.C. Paulescu” Institute of Diabetes, Nutrition, and Metabolic Diseases, approval number 114/13.07.2023, and conducted in accordance with the Declaration of Helsinki.

The dataset included clinical and demographic data from metabolic-health-center patients. Age, gender, smoking status, food, living environment (urban or rural), body composition metrics, and metabolic indicators like glucose, HbA1c, and lipid profiles were important. There was also an examination of stool samples for the presence of microorganisms.

In this study, we chose to use the minimization randomization method [14]. Minimization involves defining a mathematical function *f* (referred to as the minimization function), which assigns a numeric score to each of the two patient groups (control group, G1, and experimental group, G2). This function is unique to each clinical study and accounts for both general biological parameters of the patients (such as age and sex) and study-specific prognostic factors [14].

The inclusion criteria included the following: Our goal was to recruit a representative sample of persons aged 18–75 who consented to the study. Participants had to have no history of serious gastrointestinal illnesses, malignancies, or antibiotic usage, which can change microbiota composition. All participants had T2DM diagnosed during the past decade and were treated with metformin at 500–2000 mg/day.

The exclusion criteria included the following: Chronic inflammatory disorders unrelated to metabolic health, recent hospitalization, pregnancy, or immunosuppressive medication usage were excluded.

The biochemical analysis included the following: After an overnight fast, blood samples were analyzed using an automated biochemistry analyzer Cobas Integra 400 Plus to measure fasting blood glucose, HbA1c, lipid profiles (total cholesterol, HDL, LDL, and triglycerides), renal function markers (urea and creatinine), and inflammatory markers [15]. Also tested were liver enzymes (AST, ALT, and GGT).

The MC-780MA-N analyzer used bioelectrical impedance analysis (BIA) to measure body composition, including total body water, visceral fat, muscle mass, and metabolic age [16]. The device’s software automatically calculated basal metabolic rate (BMR), an indicator of resting metabolic rate (RMR) [17]. The device automatically determined BMR, an indicator of RMR, based on body composition measurements. To ensure accuracy, participants were instructed to fast and avoid physical activity for 12 h before the measurement.

IMD LABOR Berlin, a certified German laboratory, used a simplified quantitative or PCR-based technique for the analysis [18]. The QIIME2 pipeline identified *Bifidobacterium*, *Lactobacillus*, *Escherichia coli*, and *Candida* spp. from the data. The microbiome pH was evaluated with a biological sample pH meter, and the samples were held in sterile containers, refrigerated, transported, and analyzed within 24 h.

The statistical analysis included the following: SPSS 29 was used for statistical analysis [19]. Demographic and clinical data were summarized using descriptive statistics. As needed, chi-square tests, *t*-tests, or Mann–Whitney U tests were used to examine the correlations between categorical factors like smoking status and diet and continuous outcomes like body composition and metabolic indicators [20]. We examined metabolic markers and microbial diversity using correlation analysis. To account for age, gender, and living environment, multivariable regression models were used.

## 3. Results

Table 1 shows the cohort’s demographic, anthropometric, metabolic, and biochemical characteristics. The dataset includes 60 individuals with an average age of 64.2 years (±10.68), indicating an older demographic. Almost 43.3% are men, and 83.3% live in cities. Smoking rates are around 27%. A BMI distribution shows that the majority are overweight or obese, with an average height of 169.03 cm and weight of 82 kg. Central obesity is indicated by 102.83 cm abdominal circumference.

Metabolic and biochemical parameters show cohort health. This type 2 diabetes population has a well-balanced glucose profile with a mean HbA1c of 6.58% and a mean fasting blood glucose of 135.77 mg/dL. The lipid profile included the following: 177.2 mg/dL total cholesterol, 50.07 mg/dL HDL, and 107.14 mg/dL LDL. Triglycerides fluctuate, averaging 166.62 mg/dL, indicating lipid metabolism.

Hs-CRP (2.52 mg/L), IL-6 (3.46 pg/mL), and liver enzymes (AST, ALT, and GGT) can measure systemic inflammation and hepatic function. As evaluated by creatinine (0.775 mg/dL) and eGFR (93.92 mL/min/1.73 m^2^), most patients had maintained renal function. Gut pH was 6.57, total body water, 47.43%, and visceral fat, 10.1.

Obesity-related parameters strongly predict metabolic dysfunction in the cohort study. Overweight and obese individuals (BMI > 25 kg/m^2^) had increased fasting blood glucose (mean of 135.77 mg/dL) and HbA1c (mean of 6.58%), suggesting glucose dysregulation and insulin resistance. Obesity was linked to dyslipidemia, as those with elevated BMI and belly circumference had higher LDL cholesterol (107.14 mg/dL) and triglycerides (166.62 mg/dL).

Visceral fat gain was associated with higher hs-CRP (mean 2.52 mg/L) and IL-6 (mean 3.46 pg/mL) levels, and ALT and GGT levels were mildly elevated in obese people.

Table 2 provides a detailed overview of how patients are distributed based on the presence and type of reduced gut bacterial taxa. The table categorizes patients according to whether they exhibit a reduction in specific groups of beneficial bacteria (*Faecalibacterium prausnitzii*, *Akkermansia muciniphila*, and *Bifidobacterium* spp.) or SCFA producers.

There is a significant proportion of patients present with a decrease in butyrate-producing bacteria, which are critical for maintaining intestinal barrier integrity, regulating immune responses, and improving insulin sensitivity.

The table also indicates a cluster of patients with reductions across multiple bacterial groups; this is particularly relevant, as simultaneous deficits in *Bifidobacterium* and *Akkermansia* have been associated with systemic low-grade inflammation and metabolic endotoxemia.

Furthermore, as it can be seen in Figure 3, stratification by demographic or lifestyle variables (e.g., high stress, alcohol use, or smoking) reveals consistent correlations between unfavorable behaviors and bacterial depletion.

As per Table 3, a percentage of patients exhibited increased levels of opportunistic or pro-inflammatory bacteria.

Among them were *Clostridium* spp., *Enterobacter* spp., and *Klebsiella* spp., all associated with endotoxin production, metabolic endotoxemia, and impaired insulin signaling when present in excess.

Furthermore, there are increased levels of gas-producing or proteolytic bacteria (*Proteus* spp. and certain *Firmicutes* strains). The table and Figure 4 also reveal that increased bacterial levels were more commonly observed in patients with poor dietary habits, stress, and sedentary behavior.

Table 4 highlights the presence and diversity of fungal species detected among the patient group. A notable percentage of patients harbored detectable levels of fungi, with Candida spp. being the most commonly identified genus.

Table 5 presents the distribution of patients according to their treatment group and their self-reported daily intake of fruits and vegetables.

Among sitagliptin users, 80% reported daily use, while 77.4% of empagliflozin users did the same. The difference was not statistically significant (*p* = 1.000), indicating similar adherence patterns between the two groups.

Figure 5 compares the percentage of patients consuming fruits and vegetables daily across the two groups. As it can be seen, a smaller proportion of patients from Januavia and Empagliflozin groups, 20% and 22.6%, respectively, reported no daily consumption.

Table 6 presents the frequency of daily animal product consumption—such as meat, dairy, and eggs—among patients grouped by antidiabetic treatment. There is a high prevalence of daily animal product intake across both treatment groups, with only slight variations between them.

Figure 6 compares the percentage of patients consuming animal products daily between the two treatment groups.

Among those taking sitagliptin, 80% reported daily consumption, compared to 64.5% in the empagliflozin group. Conversely, 20% of sitagliptin users and 35.5% of Empagliflozin users reported not consuming animal products daily.

The data in Table 7 represent the comparative evolution of blood glucose levels between visits in patients treated with sitagliptin. The distribution of blood glucose values was non-parametric at the second-visit measurement according to the Shapiro–Wilk test (*p* < 0.05).

Table 8 shows that there is a significant reduction in glycemia over time, with median values decreasing from 131 mg/dL (IQR 110–145) at the initial visit to 114 mg/dL (IQR 98–130) at follow-up.

This decline was statistically significant (*p* = 0.025), as confirmed by the Wilcoxon signed-rank test. This indicates that empagliflozin effectively improved short-term glycemic control.

Figure 7 and Figure 8 compare both treatment groups in terms of blood glucose for the 1st and 2nd visit.

The differences between visits were significant according to the Wilcoxon test (*p* = 0.046), showing a significant decrease in blood glucose values from the initial value (median = 132, IQR = 111–146) compared to the final value (median = 123, IQR = 101–135). The observed difference was statistically significant (median = −2, IQR = −18.85 to 3). For the illustration of the quantitative values distributions in the box-plot graphs, the IBM SPSS Statistics software illustrates any values that are below the 1st quartile (25th percentile) – 1.5* interquartile range or above the 3rd quartile (75th percentile) + 1.5* interquartile range, as outliers represented by circles in the graph. As for values that are below the 1st quartile (25th percentile) – 3* interquartile range or above the 3rd quartile (75th percentile) + 3* interquartile range, the software represents the values as extreme outliers represented by asterisk symbols in the graph.

A clear reduction in the median glucose level is observed from the 1st visit to the 2nd visit, indicating improved glycemic control over time. Both boxplots display a similar interquartile range (IQR). Thus, variability among patients remained relatively stable. However, since the overall downward shift in values, along with the presence of a few outliers above 200 mg/dL in both visits is noted, there is a general trend toward lower glucose levels. This has to take in consideration the individual variability in treatment response.

The data in Table 9 and Table 10 represent the longitudinal comparison of HbA1c values across visits in patients treated with sitagliptin and empagliflozin, respectively.

The distribution of HbA1c values was non-parametric at both measurements according to the Shapiro–Wilk test (*p* < 0.05).

The results show a significant reduction in HbA1c from a median value of 6.5% at baseline to 6.2% at the 2nd visit, with this change reaching statistical significance (*p* = 0.001). Thus, empagliflozin managed to stabilize glucose metabolism through its insulin-independent mechanism of action.

The differences between visits were statistically significant according to the Wilcoxon test (*p* = 0.049), showing a significant decrease in HbA1c values from the initial value (median = 6.3, IQR = 6–6.75) compared to the final value (median = 6.2, IQR = 6–6.6). The observed difference was significant (median = −0.1, IQR = −0.35 to 0).

As per Table 11, a visible reduction in the median HbA1c—from approximately 6.5% to around 6.2%—suggests a modest but meaningful improvement in long-term glycemic control. The interquartile range appears slightly tighter at the 2nd visit. This indicates less variability in patient responses post-treatment.

Both groups represented in Figure 9 and Figure 10 demonstrated a reduction in HbA1c; however, the decrease was more pronounced in the empagliflozin group. Specifically, patients in this group showed a statistically significant reduction from 6.5% to 6.2% (*p* = 0.001), while those in the sitagliptin group experienced a more modest decrease from 6.3% to 6.2% (*p* = 0.049).

Table 12 shows that there is a slight improvement in microbial diversity following treatment with sitagliptin, as indicated by a decrease in the prevalence of key bacterial deficiencies between the initial and follow-up visits.

There is a slight improvement in microbial diversity following treatment with sitagliptin, as indicated by a decrease in the prevalence of key bacterial deficiencies between the initial and follow-up visits. Specifically, reductions in beneficial bacteria such as *Bifidobacterium* spp. and *Lactobacillus* spp. were less frequent at the 2nd visit. Table 13 shows the outcome in patients. 

There is a modest improvement in the overall percentage of patients that had decreased levels of *Bifidobacterium* spp. and *Lactobacillus* spp. For instance, the prevalence of patients with reduced *Bifidobacterium* spp. slightly declined, and similar patterns were noted for other key taxa, although the changes were not statistically significant.

Figure 11 shows that there is an improvement in microbial balance as the total prevalence of reduced bacteria decreased from 83.3% to 70%.

*Escherichia coli* deficiencies decreased significantly (from 16.7% to 6.7%), while *Bifidobacterium* spp. and *Lactobacillus* spp. also showed modest improvements. The prevalence of *Enterococcus* spp. remained unchanged.

Table 14 and Table 15 show a longitudinal comparison of the presence and type of increased bacteria in patients treated with sitagliptin.

Opportunistic species such as *Escherichia coli* and *Clostridium* spp., Beta h. Streptococcus, and Alpha h. Streptococcus were reduced following treatment with sitagliptin.

Overall, there is an improvement in microbial balance as the presence of elevated bacteria decreased from 86.7% to 73.3%.

Figure 12 shows that *Escherichia coli* decreased from 60% to 46.7% and *Alpha hemolytic Streptococcus* from 43.3% to 30%.

Slight increases were observed in *Serratia* spp., *Citrobacter* spp., and *Lactobacillus* spp., though these remained low in prevalence.

As per Figure 13, an overall decrease is also observed in the total presence of elevated bacterial populations, dropping from 83.9% to 58.1%.

Specific reductions are particularly evident in *Escherichia coli* (from 61.3% to 38.7%) and *Alpha hemolytic Streptococcus* (from 48.4% to 16.1%). Additional decreases are noted in *E. coli Biovare*, *Klebsiella* spp., and *Enterobacteriaceae*, while some species such as *Pseudomonas* spp. and *Citrobacter* spp. appeared at low levels during the follow-up.

Table 16 presents the distribution of patients across the two treatment groups alongside the observed rate of decline in the frequency of specific bacterial species in the gut microbiome over time.

Both treatment groups experienced a reduction in the prevalence of dysbiotic bacterial taxa, though the patterns differed slightly between them. Patients treated with empagliflozin exhibited a more substantial decrease in the frequency of pro-inflammatory or opportunistic species (*Escherichia coli* and *Streptococcus* spp.).

Conversely, the sitagliptin group also demonstrated a reduction in harmful bacterial species, though the changes were more modest. In particular, patients on sitagliptin showed a slower but consistent decline in both increased and decreased microbial populations. While both drugs were associated with improvements in glycemic control, the observed microbiome shifts imply that Empagliflozin may exert a stronger modulatory effect on gut microbial composition.

## 4. Discussion

In patients with type 2 diabetes mellitus who were already getting treatment with metformin and were also undergoing treatment with sitagliptin or empagliflozin, our research offers a light on the interaction between the composition of the gut microbiota and the regulation of glycemic levels. There were significant disparities between the two medicines in terms of their impact on the microbiota in the gut, despite the fact that both treatments shown good benefits on metabolic outcomes.

Treatment with sitagliptin resulted in a moderate improvement in the balance of microorganisms in the gut. The percentage of patients who had lower levels of helpful bacteria (*Bifidobacterium* spp. and *Lactobacillus* spp.) dropped from 83.3% to 70% over the course of the study. Furthermore, the incidence of increasing pathogenic taxa, such as *Escherichia coli* and *Clostridium* spp., decreased from 86.7% to 73.3%. This was a significant decrease.

The magnitude of these microbial alterations, on the other hand, remained quite minor. It is believed that the therapeutic mechanism of sitagliptin primarily targets glucose metabolism, with secondary effects on gut microbiota perhaps being mediated by enhanced metabolic control and reduced low-grade inflammation [20,21].

Empagliflozin, on the other hand, was shown to have a more significant impact on both the glycemic indices and the immune system profiles of the gut. The proportion of patients with reduced *Lactobacillus* spp. significantly declined (from 54.8% to 29% with *p* = 0.021), and there was a pronounced reduction in the presence of pro-inflammatory bacteria (*Escherichia coli* from 61.3% to 38.7% with *p* = 0.065 and *Alpha-hemolytic Streptococcus* from 48.4% to 16.1% with *p* = 0.013). Furthermore, empagliflozin was associated with a significant decline in fungal overgrowth (*Candida* spp., *p* = 0.034), a finding not observed with sitagliptin. These effects may be attributed to empagliflozin’s SGLT2 inhibition, promoting glycosuria [22,23]. This reduces systemic glucose levels and indirectly modulates gut microbial and mycobiome environments [24].

From a metabolic perspective, empagliflozin yielded superior glycemic control, with HbA1c levels decreasing from 6.5% to 6.2% (*p* = 0.001), compared to a smaller reduction from 6.3% to 6.2% with sitagliptin (*p* = 0.049). Similarly, fasting glucose levels declined more significantly in the empagliflozin group.

The studies found in the literature also suggest that empagliflozin’s multifaceted mechanism of action may offer dual benefits: enhancing glucose control while actively promoting gut microbial rebalancing [25,26]. Conversely, research shows that sitagliptin appears to exert a milder, more gradual influence on microbiota composition, likely secondary to its metabolic effects [27,28].

Our findings are consistent with previous studies that support the integration of microbiota-focused strategies in T2DM management and suggest that SGLT2 inhibitors like empagliflozin may hold a superior therapeutic profile for patients where gut dysbiosis and metabolic syndrome coexist [29].

## 5. Limitations

One of the primary limitations of this study is the lack of direct microbiota profiling through molecular techniques. Instead, bacterial presence and imbalances were inferred through microbiological culture methods. Many commensal and beneficial bacterial species are anaerobic and not easily cultivable using standard laboratory techniques, potentially underestimating microbial richness and functional interactions.

Additionally, dietary intake and lifestyle factors were self-reported and not controlled through standardized interventions, which introduces the potential for bias and variability.

Finally, the relatively short follow-up period may not fully capture long-term microbiota shifts or their clinical consequences, highlighting the need for extended longitudinal studies with larger cohorts and mechanistic microbiome analyses.

## 6. Conclusions

This study highlights the complex relationship between gut microbiota makeup and metabolic management in type 2 diabetes patients receiving pharmacological treatment. Both sitagliptin and empagliflozin improved glycemic indices and microbiome profiles; however, the extent and nature of these changes differed.

Sitagliptin reduced fasting glucose and HbA1c and decreased helpful microorganisms and increased harmful species. The gradual restoration of microbial equilibrium was shown by decreases in *Escherichia coli*, *Bifidobacterium*, and *Lactobacillus*. Although the microbial and metabolic alterations were slow, sitagliptin may indirectly affect the gut microbiome by stabilizing metabolism and reducing systemic inflammation.

Glycemic control and microbial modulation were stronger in empagliflozin-treated subjects. HbA1c and fasting glucose decreased significantly, as did pro-inflammatory microorganisms including *Escherichia coli* and *Alpha haemolytic Streptococcus*. Empagliflozin also reduced high bacterial populations and accentuated gut microbial balance. Both medications improve metabolic and microbial health in T2DM, but empagliflozin appears to have a greater impact on glycemic outcomes and gut microbiota restoration.

## Figures and Tables

**Figure 1 metabolites-15-00411-f001:**
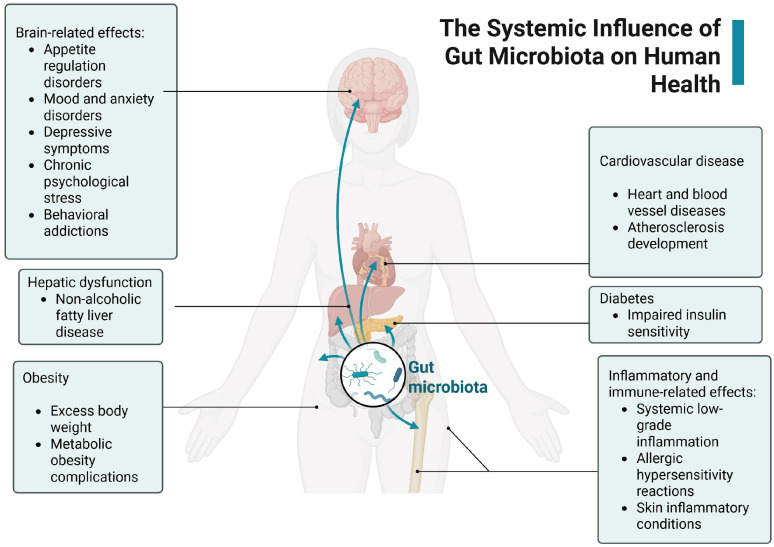
Systemic effects of gut microbiota across multiple physiological and organ systems, including the brain, liver, cardiovascular system, adipose tissue, and gastrointestinal tract. The diagram highlights the bidirectional communication pathways between the gut microbiota and host metabolic, immune, and endocrine functions, emphasizing the role of microbial dysbiosis in contributing to chronic inflammation, metabolic dysregulation, and cardiometabolic disease. Created with Biorender [4].

**Figure 2 metabolites-15-00411-f002:**
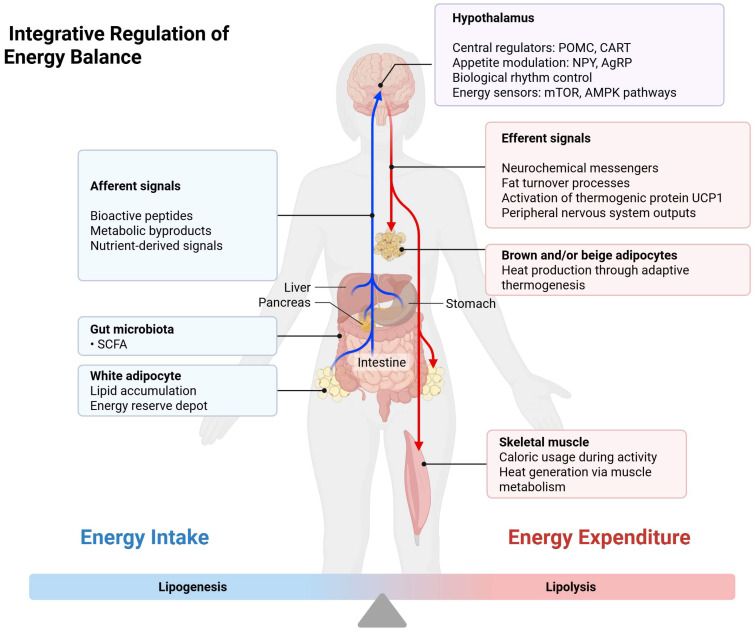
Interplay between the gut microbiota, the central nervous system, and peripheral tissues in regulating energy homeostasis. Microbial metabolites, such as short-chain fatty acids, interact with the gut–brain axis to regulate insulin sensitivity, glucose metabolism, and appetite control, providing a mechanistic background for the study’s investigation into diabetes-related microbiota changes. Created with Biorender [4].

**Figure 3 metabolites-15-00411-f003:**
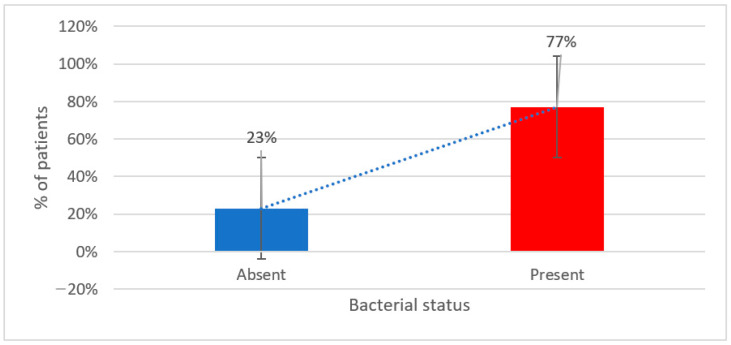
Prevalence of reduced gut bacteria among study participants.

**Figure 4 metabolites-15-00411-f004:**
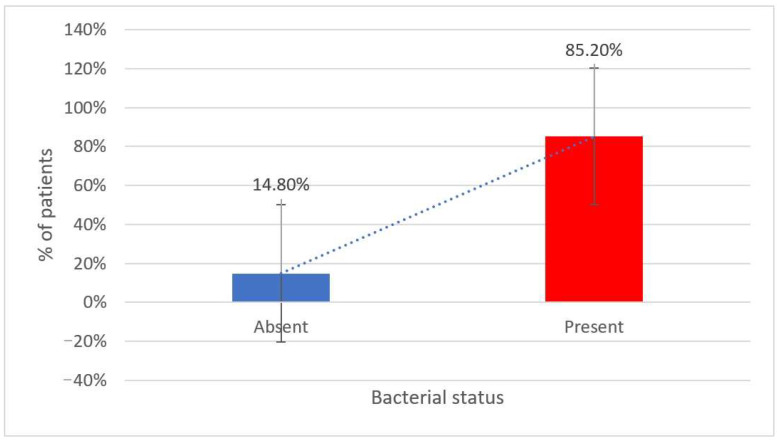
Prevalence of increased gut bacteria among study participants.

**Figure 5 metabolites-15-00411-f005:**
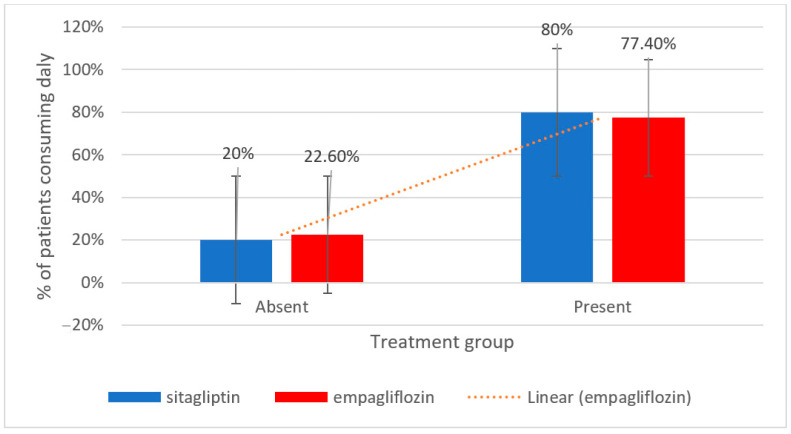
Daily fruit and vegetable consumption in patients treated with sitagliptin vs. empagliflozin.

**Figure 6 metabolites-15-00411-f006:**
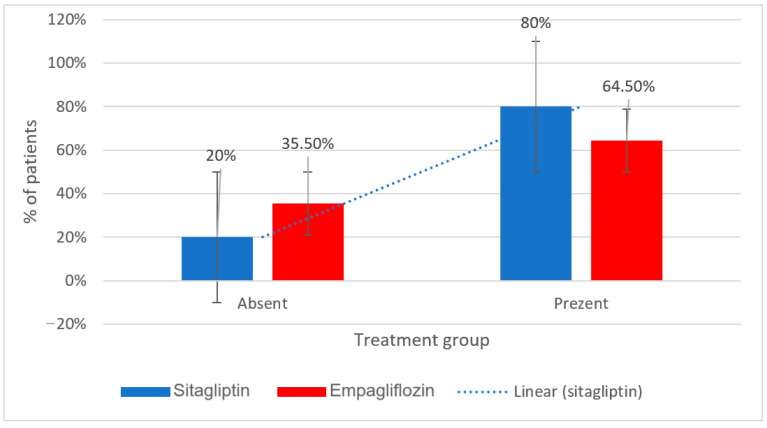
Daily animal product consumption in patients treated with sitagliptin vs. empagliflozin.

**Figure 7 metabolites-15-00411-f007:**
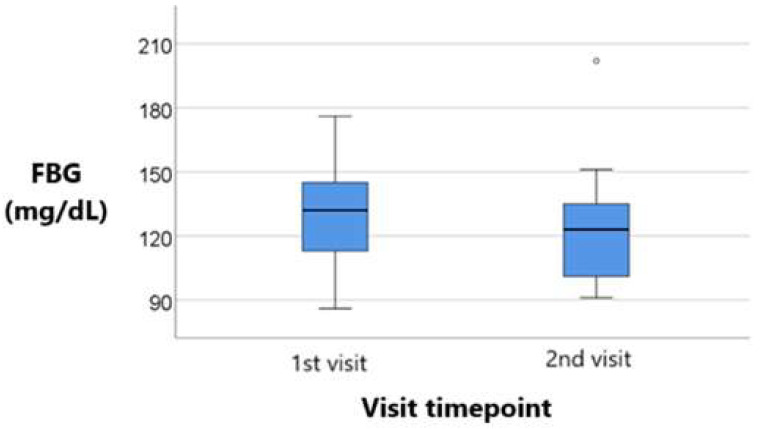
Longitudinal comparison of blood glucose values across visits in patients treated with sitagliptin. Median, interquartile range (IQR), and individual variability are shown, with statistical significance assessed using the Wilcoxon signed-rank test. This figure demonstrates the treatment’s glycemic effects over time.

**Figure 8 metabolites-15-00411-f008:**
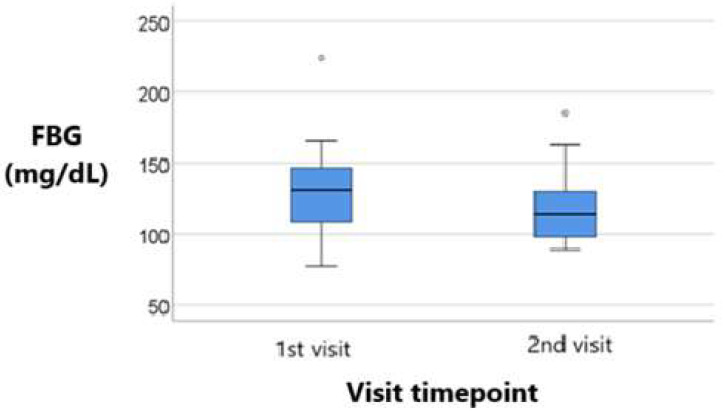
Longitudinal comparison of blood glucose values across visits in patients treated with empagliflozin. Median values and IQR are presented, highlighting the treatment’s impact on short-term glycemic control, with statistical significance reported.

**Figure 9 metabolites-15-00411-f009:**
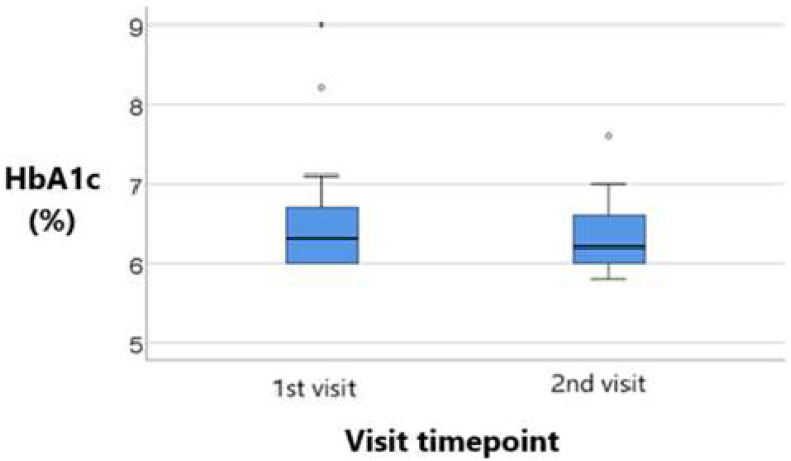
Longitudinal comparison of HbA1C values across visits in patients treated with sitagliptin. Median, IQR, and individual variability are displayed, illustrating the modest but statistically significant improvement in long-term glycemic control with DPP-4 inhibitor therapy.

**Figure 10 metabolites-15-00411-f010:**
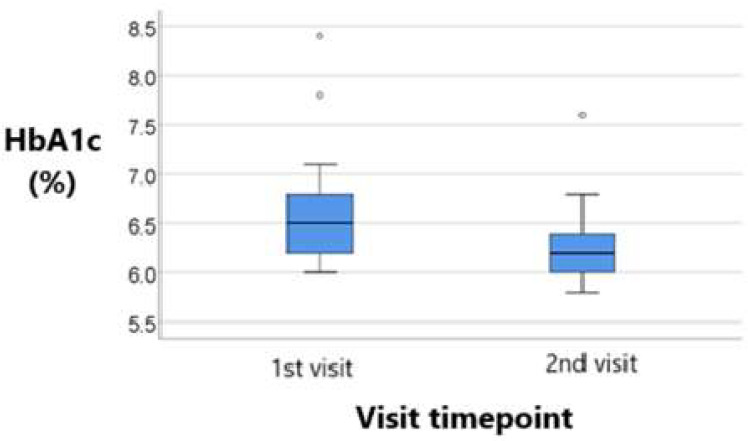
Longitudinal comparison of HbA1C values across visits in patients treated with empagliflozin. Greater reduction in HbA1c with SGLT-2 inhibitor therapy, including variability and significance testing (Wilcoxon signed-rank test).

**Figure 11 metabolites-15-00411-f011:**
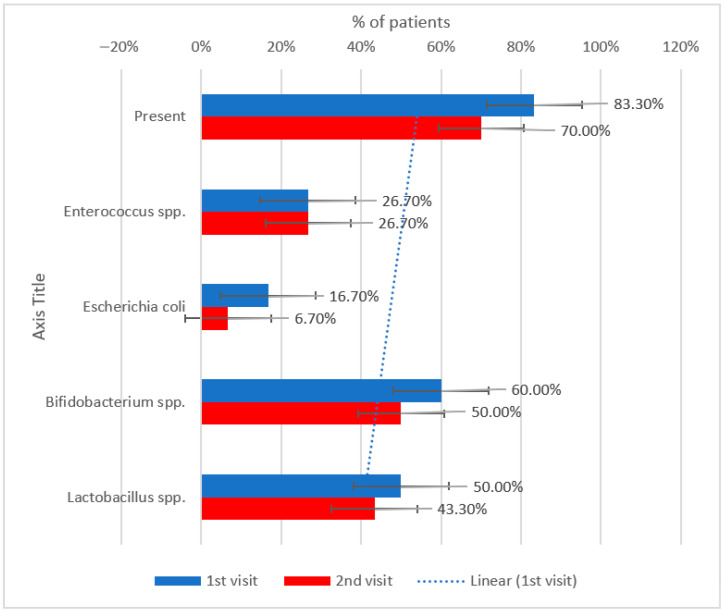
Evolution of reduced gut bacterial presence between the initial visit and the second visit in patients treated with sitagliptin.

**Figure 12 metabolites-15-00411-f012:**
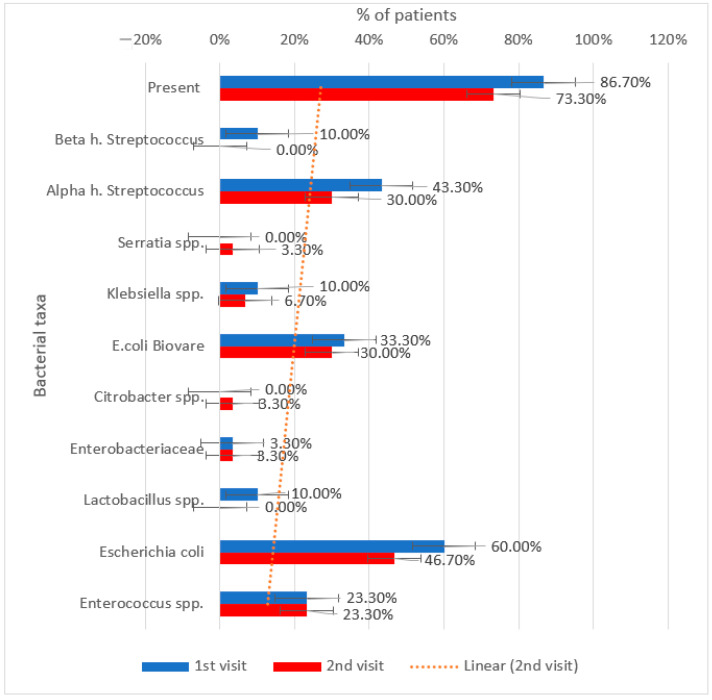
Evolution of increased gut bacterial presence between the initial visit and the second visit in patients treated with sitagliptin.

**Figure 13 metabolites-15-00411-f013:**
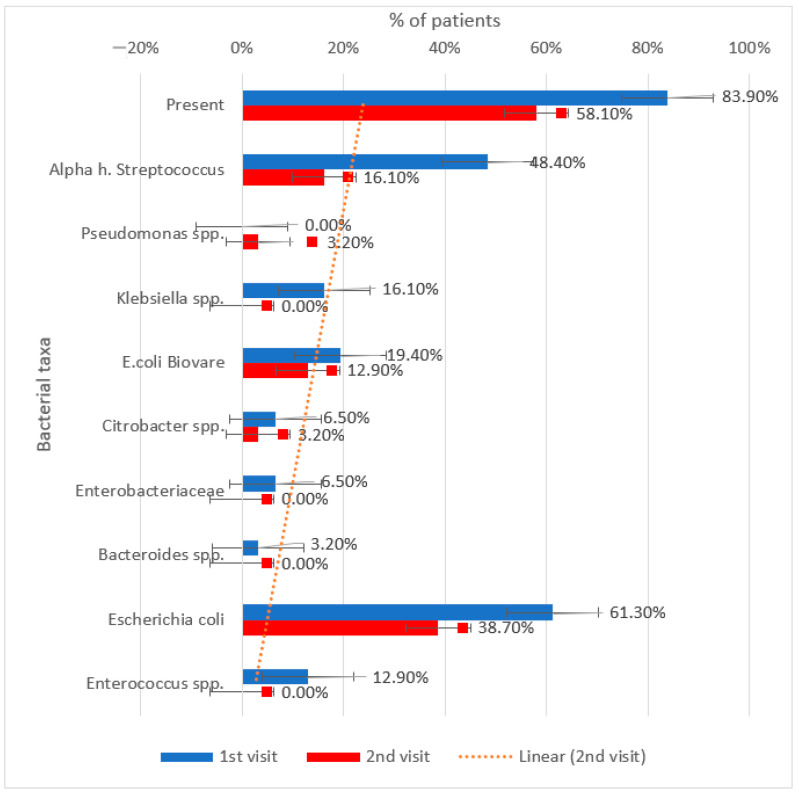
Evolution of increased gut bacterial presence between the initial visit and the second visit in patients treated with empagliflozin.

**Table 1 metabolites-15-00411-t001:** Population characteristics.

Variable	No.	Mean or %	Std. Deviation *	Variance *
Age	60	64.2	10.6	114.040
Male	30	43.3%	N/A	N/A
Urban environment	30	83.3%	N/A	N/A
Non-smoker	30	73.3%	N/A	N/A
Height (cm)	60	169.033	8.961	80.299
Weight (kg)	60	82.000	14.281	203.940
Abdominal circumference (cm)	60	102.833	8.851	78.339
BMI	60	28.933	3.361	11.298
Normal weight	8	13.3	0.4	0.16
Overweight	11	18.3	1.236	1527
Class I	19	31.6	1.216	1.48
Class II	22	36.6	0.852	0.726
Hemoglobin (Hb) (g/dL)	60	14.233	1.221	1.492
Leukocytes (10^3^/µL)	60	7.545	1.328	1.764
Platelets (10^3^/µL)	60	267.900	83.364	6949.557
HbA1C (%)	60	6.583	0.596	0.355
Blood glucose (mg/dL)	60	135.767	26.627	708.979
Urea (mg/dL)	60	30.100	6.273	39.357
Creatinine (mg/dL)	60	0.775	0.167	0.028
Uric acid (mg/dL)	60	5.097	1.044	1.090
Total cholesterol (mg/dL)	60	177.200	45.736	2091.811
HDL (mg/dL)	60	50.071	13.280	176.350
LDL (mg/dL)	60	107.138	43.276	1872.783
Triglycerides (mg/dL)	60	166.625	94.171	8868.190
AST (TGO) (U/L)	60	21.433	5.536	30.646
ALT (TGP) (U/L)	60	27.833	11.719	137.339
GGT (U/L)	60	25.241	12.724	161.907
FT4 (pmol/L)	60	16.783	2.574	6.626
TSH (µIU/mL)	60	2.287	1.090	1.188
CRP (mg/L)	60	2.857	3.416	11.667
hs-CRP (mg/L)	60	2.525	3.084	9.508
IL-6 (pg/mL)	60	3.459	2.174	4.728
UACR	60	928.157	2084.693	4,345,944.302
eGFR (mL/min/1.73 m^2^)	60	93.924	13.981	195.466
Microbiome pH	60	6.567	0.642	0.412
Total body water (%)	60	47.430	5.350	28.619
Visceral fat level	60	10.100	3.969	15.757
Basal metabolic rate (kcal)	60	1708.700	310.164	96,202.010
Metabolic age (years)	60	57.267	12.331	152.062
Muscle mass (kg)	60	55.113	10.076	101.519
Muscle mass percentage (%)	60	66.067	6.502	42.278

* Std. deviation and variance use N rather than N-1 in denominators.

**Table 2 metabolites-15-00411-t002:** Patients distribution according to presence and type of reduced gut bacteria.

Decreased Bacteria	Number	Percentage
Absent	14	23%
Present	47	77%
Type	Number	Percentage (Total)
*Enterococcus* spp.	16	26.2%
*Escherichia coli*	6	9.8%
*Bifidobacterium* spp.	36	59%
*Lactobacillus* spp.	32	52.5%

**Table 3 metabolites-15-00411-t003:** Distribution of patients based on the presence and specific types of increased bacterial populations detected in the analyzed samples.

Increased Bacteria	Number	Percentage
Absent	9	14.8%
Present	52	85.2%
Tip bacterii	Number	Percentage (Total)
*Beta h. Streptococcus*	3	4.9%
*Alpha h. Streptococcus*	28	45.9%
*Klebsiella* spp.	8	13.1%
*E. coli Biovare*	16	26.2%
*Citrobacter* spp.	2	3.3%
*Enterobacteriaceae*	3	4.9%
*Lactobacillus* spp.	3	4.9%
*Bacteroides* spp.	1	1.6%
*Escherichia coli*	37	60.7%
*Enterococcus* spp.	11	18%

**Table 4 metabolites-15-00411-t004:** Distribution by presence and type of fungi.

Fungi	Number	Percentage
Absent	31	50.8%
Present	30	49.2%
Tip fungi	Number	Percentage (Total)
*Candida* spp.	22	36.1%
*Candida albicans*	8	13.1%
*Geotrichum* spp.	5	8.2%
*Filamentous fungus*	2	3.3%

**Table 5 metabolites-15-00411-t005:** Treatment group and daily fruit/vegetable consumption.

Daily Use	Sitagliptin		Empagliflozin		*p* *
	No.	Percentage	No.	Percentage	
Absent	6	20%	7	22.6%	1.000
Present	24	80%	24	77.4%	

* Fisher’s exact test.

**Table 6 metabolites-15-00411-t006:** Treatment group and daily animal product consumption.

Daily Use	Sitagliptin		Empagliflozin		*p* *
	No.	Percentage	Nr.	Percentage	
Absent	6	20%	11	35.5%	0.255
Present	24	80%	20	64.5%	

* Fisher’s exact test.

**Table 7 metabolites-15-00411-t007:** Longitudinal comparison of blood glucose values across visits in patients treated with sitagliptin.

Blood Glucose/Visit	Mode ± SD	Median (IQR)	*p* *
1st visit (*p* = 0.790 **)	128.93 ± 23.98	132 (111–146)	0.046
2nd visit (*p* = 0.004 **)	120.59 ± 24.11	123 (101–135)	
Difference	−8.33 ± 21.04	−2 (−18.85–3)	-

* Related-samples Wilcoxon signed rank test, ** Shapiro–Wilk test.

**Table 8 metabolites-15-00411-t008:** Longitudinal comparison of blood glucose values across visits in patients treated with empagliflozin.

Blood Glucose/Visit	Mode ± SD	Median (IQR)	*p* *
1st visit (*p* = 0.052 **)	130.06 ± 28.54	131 (107–147)	0.025
2nd visit (*p* = 0.007 **)	117 ± 21.4	114 (98–131)	
Difference	−13.06 ± 27.65	−10 (−23–10)	-

* Related-samples Wilcoxon signed rank test, ** Shapiro–Wilk test.

**Table 9 metabolites-15-00411-t009:** Longitudinal comparison of HbA1C values across visits in patients treated with sitagliptin.

HbA1c/Visit	Mode ± SD	Median (IQR)	*p* *
1st visit (*p* < 0.001 **)	6.51 ± 0.69	6.3 (6–6.75)	0.049
2nd visit (*p* = 0.008 **)	6.33 ± 0.4	6.2 (6–6.6)	
Difference	−0.172 ± 0.445	−0.1 (−0.35–0)	-

* Related-samples Wilcoxon signed rank test, ** Shapiro–Wilk test.

**Table 10 metabolites-15-00411-t010:** Longitudinal comparison of HbA1C values across visits in patients treated with empagliflozin.

HbA1c/Visit	Mode ± SD	Median (IQR)	*p* *
1st visit (*p* < 0.001 **)	6.56 ± 0.52	6.5 (6.2-6.8)	0.001
2nd visit (*p* < 0.001 **)	6.25 ± 0.36	6.2 (6-6.4)	
Difference	−0.316 ± 0.45	−0.3 (−0.5–-0.2)	-

* Related-samples Wilcoxon signed rank test, ** Shapiro–Wilk test.

**Table 11 metabolites-15-00411-t011:** Comparison of HbA1c changes over time between treatment groups.

Group	Mone ± SD	Median (IQR)	Average Range	*p* *
sitagliptin (*p* = 0.016 **)	−0.17 ± 0.44	−0.1 (−0.35–0)	35.22	0.042
empagliflozin (*p* = 0.003 **)	−0.32 ± 0.45	−0.3 (−0.5–−0.2)	26.08	

* Mann–Whitney U test, ** Shapiro–Wilk test.

**Table 12 metabolites-15-00411-t012:** Longitudinal comparison of the presence and type of reduced bacteria in patients treated with sitagliptin.

Decreased Bacteria (No., %)	1st Visit	2nd Visit	*p* *
Present	25 (83.3%)	21 (70%)	0.219
*Enterococcus* spp.	8 (26.7%)	8 (26.7%)	1.000
*Escherichia coli*	5 (16.7%)	2 (6.7%)	0.375
*Bifidobacterium* spp.	18 (60%)	15 (50%)	0.453
*Lactobacillus* spp.	15 (50%)	13 (43.3%)	0.727

* Related-samples McNemar test.

**Table 13 metabolites-15-00411-t013:** Longitudinal comparison of the presence and type of reduced bacteria in patients treated with empagliflozin.

Decreased bacteria (No., %)	1st visit	2nd visit	*p* *
Present	22 (71%)	19 (61.3%)	0.453
*Enterococcus* spp.	8 (25.8%)	10 (32.3%)	0.727
*Escherichia coli*	1 (3.2%)	3 (9.7%)	0.500
*Bifidobacterium* spp.	18 (58.1%)	13 (41.9%)	0.227
*Lactobacillus* spp.	17 (54.8%)	9 (29%)	0.021

* Related-samples McNemar test.

**Table 14 metabolites-15-00411-t014:** Longitudinal comparison of the presence and type of increased bacteria in patients treated with sitagliptin.

Increased Bacteria (No., %)	1st Visit	2nd Visit	*p* *
Present	26 (86.7%)	22 (73.3%)	0.125
*Beta h. Streptococcus*	3 (10%)	0 (0%)	0.250
*Alpha h. Streptococcus*	13 (43.3%)	9 (30%)	0.289
*Serratia* spp.	0 (0%)	1 (3.3%)	1.000
*Klebsiella* spp.	3 (10%)	2 (6.7%)	1.000
*E. coli Biovare*	10 (33.3%)	9 (30%)	1.000
*Citrobacter* spp.	0 (0%)	1 (3.3%)	1.000
*Enterobacteriaceae*	1 (3.3%)	1 (3.3%)	1.000
*Lactobacillus* spp.	3 (10%)	0 (0%)	0.250
*Escherichia coli*	18 (60%)	14 (46.7%)	0.125
*Enterococcus* spp.	7 (23.3%)	7 (23.3%)	1.000

* related-samples McNemar test.

**Table 15 metabolites-15-00411-t015:** Longitudinal comparison of the presence and type of increased bacteria in patients treated with empagliflozin.

Increased Bacteria (No., %)	1st Visit	2nd Visit	*p* *
Present	26 (83.9%)	18 (58.1%)	0.039
*Alpha h. Streptococcus*	15 (48.4%)	5 (16.1%)	0.013
*Pseudomonas* spp.	0 (0%)	1 (3.2%)	1.000
*Klebsiella* spp.	5 (16.1%)	0 (0%)	0.063
*E. coli Biovare*	6 (19.4%)	4 (12.9%)	0.688
*Citrobacter* spp.	2 (6.5%)	1 (3.2%)	1.000
*Enterobacteriaceae*	2 (6.5%)	0 (0%)	0.500
*Bacteroides* spp.	1 (3.2%)	0 (0%)	1.000
*Escherichia coli*	19 (61.3%)	12 (38.7%)	0.065
*Enterococcus* spp.	4 (12.9%)	0 (0%)	0.125

* related-samples McNemar test.

**Table 16 metabolites-15-00411-t016:** Comparison of bacteria absent over time between groups.

Group/Species Absent Over Time	Sitagliptin		Empagliflozin		*p* *
	Nr.	%	Nr.	%	
*Lactobacillus* spp.	5	16.7%	9	29%	0.363
*Bacterii crescute (general)*	4	13.3%	10	32.3%	0.127
*Alpha h. Streptococcus*	6	20%	12	38.7%	0.161
*Klebsiella* spp.	2	6.7%	5	16.1%	0.425
*Escherichia coli*	4	13.3%	9	29%	0.211
*Fungi (general)*	3	10%	10	32.3%	0.034 **
*Candida* spp.	5	16.7%	10	32.3%	0.235

* Fisher’s exact test, ** Pearson chi-square test.

## Data Availability

The raw data supporting the conclusions of this article will be made available by the authors on request.

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
