# Peer review of "SGLT-2 Inhibitors and Metabolic Outcomes: A Primary Data Study Exploring the Microbiota–Diabetes Connection"

_metabolites, 2025, doi:10.3390/metabo15060411_

Round 1
Reviewer 1 Report
Comments and Suggestions for Authors
Abstract: In the sentence, “Microbiota analysis revealed reductions in the prevalence of reduced beneficial bacteria” (Line 31-32), does the treatment increase beneficial bacteria or reduce the reduction? That point is to be made clear in the abstract
Line 34: The term “rebalancing” is confusing. Specify whether it was an increase or a decrease
Introduction:
- Introduction contains some broad topics, which may dilute the main focus of the paper
- Line 44: "This microbial ecosystem plays a and metabolic regulation [1]." The sentence is incomplete
- The final paragraph introduces metformin but lacks a transition to SGLT-2 inhibitors. The title mentions SGLT-2 inhibitors, but they appear only very late and briefly.
Materials and Methods: Too brief and simple for a scientific publication
- : Line 107 “stool samples were analysed for bacteria.” Incomplete. How were the samples collected and processed? What are the techniques employed for their characterisation
- Line 125-126: Participants were “instructed to” rather than “encouraged”
- Most of the observations present in the results do not appear in the materials and methods section
Results
Substantial revision is needed in the Results section. Specifically:
- The graphs currently lack error bars (standard deviation or standard error), which are essential for interpreting data variability and significance.
- Figures 1 to 6 require redrawing to improve clarity, formatting, and visual consistency.
- Several graphs are missing axis titles, which compromises interpretability and should be corrected.
- The number of individual graphs is excessive; consolidating related figures into composite plates would improve visual coherence and presentation.
- The number of tables is also high. Consider merging those with overlapping content or relocating less critical data to the supplementary material to enhance focus and readability.
Discussion: is too brief for an article
- Lack of clarity and consistency in comparing sitagliptin and empagliflozin.
- Remove fragmented or incomplete sentences (the original ending was abrupt).
- Correct grammar and phrasing (e.g., "Our findings align with the ones found in the literature" → "Our findings are consistent with previous studies").
- Maintain a professional tone and integrate citations fluidly.
For more specific corrections, see the attached documents

The language should be revised to meet the standards of clarity and precision expected in scientific publications.
Reviewer 2 Report
Comments and Suggestions for Authors
Dear Authors,
Thank you for the opportunity to review your manuscript titled “SGLT-2 Inhibitors and Metabolic Outcomes: A Primary Data Study Exploring the Microbiota-Diabetes Connection.” Your study addresses an important and increasingly relevant topic, investigating the relationship between antidiabetic therapy and gut microbiota composition in individuals with type 2 diabetes mellitus. The manuscript is generally well-structured and supported by data. I appreciate the effort that went into the design and analysis.
1. The introduction is informative and provides sufficient background for understanding the significance of the topic. The connection between gut microbiota and metabolic health is well explained. However, the introduction could benefit from a brief paragraph that more clearly emphasizes what distinguishes this study from previous work, especially in terms of study population, methodology, or comparison of drugs.
2. The study design is appropriate, and the inclusion and exclusion criteria are clearly stated. The use of clinical, biochemical, and microbiological parameters strengthens the findings. That said, please clarify the randomization process in more detail. Was it simple or stratified randomization? Was allocation concealment applied?
3. The methods section is generally adequate. However, while the manuscript notes that a simplified PCR-based method was used for microbiota analysis, it should be more explicitly stated that next-generation sequencing (e.g., 16S rRNA or metagenomic approaches) was not used. This limitation is briefly mentioned in the Discussion, but I recommend also noting it in the Methods section to ensure transparency.
4. An important point: while the manuscript includes a statement on informed consent, I could not locate a sentence indicating that the study was approved by an ethics committee. Please provide the name of the institutional review board or ethics committee and the corresponding approval number.
5. The results are presented clearly, and the tables and figures are informative. Some figure labels and axis titles are too small and should be adjusted for better readability. Additionally, the figures would benefit from more detailed legends to explain symbols, especially in box plots.
6. The discussion and conclusion sections are consistent with the results. You have clearly explained the differences between sitagliptin and empagliflozin in terms of both glycemic control and microbiota modulation. Still, the discussion could be slightly expanded by considering how these findings might influence future clinical approaches or patient selection for SGLT-2 inhibitors.
In conclusion, this is a valuable and timely study. The findings contribute to the growing field of microbiota research in metabolic diseases. I recommend minor revision to address the points mentioned above, particularly the addition of the ethics approval statement and minor improvements in figure clarity.
Best regards
Reviewer 3 Report
Comments and Suggestions for Authors
The results comparing the effects of sitagliptin and empagfiflozin have clearly taken a lot of time and effort, but have yielded very little evidence of significant changes in the microbiome between these two treatments. Furthermore the way the results have been presented showing single average percentage changes without any indications of errors Figure 11, 10, 12 table 12,14 is hard to interpret, without some indication of the variablilty between patients and between visits.
Round 2
Reviewer 3 Report
Comments and Suggestions for Authors
The paper has been improved by inclusion of statistics. These make the findings more comprehensible. I think the readability overall would be improved by reduction in the numbers of tables particularly where comparisons of non significant effects of the drugs on types of bacteria are concerned. Perhaps these should be displaced to a supplementary file. However I leave this to the choice of the authors and publishers.
Author Response
Dear Reviewer,
Thank you for your thoughtful feedback and are grateful for the positive assessment regarding the inclusion of statistical analyses and their role in enhancing the comprehensibility of our findings.
With regard to the recommendation about reducing the number of tables, particularly those presenting comparisons of non-significant effects of the drugs on bacterial taxa, we have carefully considered this suggestion. However, we respectfully prefer to retain the current format and table placement within the main manuscript. Our rationale is that the tables provide essential context for interpreting the nuances of microbiota dynamics in response to treatment. The inclusion of these data within the main text ensures transparency and allows readers to appreciate both the significant and non-significant trends observed across taxa.
We believe this level of detail supports the integrity and reproducibility of our findings, especially in a research area where subtle microbial changes can have important biological implications, even if not all are statistically significant. Nevertheless, we are open to the editorial team’s guidance should they consider it more appropriate to relocate selected tables to a supplementary file.
Thank you once again for your valuable input.
Dr. Nicoleta Mindrescu